# Neutrophil degranulation, NETosis and platelet degranulation pathway genes are co-induced in whole blood up to six months before tuberculosis diagnosis

Stuart Meier[1,2,3,4¤a], James A. Seddon[4,5,6], Elizna Maasdorp[1,2,3,4,7], Léanie Kleynhans[1,2,3], Nelita du Plessis[1,2,3], Andre G. Loxton[1,2,3], Stephanus T. Malherbe[1,2,3], Daniel E. Zak[8], Ethan Thompson[8], Fergal J. Duffy[8], Stefan H. E. Kaufmann[9,10,11], Tom H. M. Ottenhoff[12], Thomas J. Scriba[13], Sara Suliman [13¤b], Jayne S. Sutherland[14], Jill Winter[15], Helena Kuivaniemi [1,2,3], Gerhard Walzl[1,2,3,4], Gerard Tromp [1,2,3,4,7] *, GC6-74 Consortium[¶], Catalysis TB Biomarkers Consortium[¶]

1 Division of Molecular Biology and Human Genetics, Department of Biomedical Sciences, Stellenbosch University, Cape Town, South Africa, 2 DSI–NRF Centre of Excellence for Biomedical Tuberculosis Research, Stellenbosch University, Cape Town, South Africa, 3 South African Medical Research Council Centre for Tuberculosis Research, Stellenbosch University, Cape Town, South Africa, 4 South African Tuberculosis Bioinformatics Initiative, Stellenbosch University, Cape Town, South Africa, 5 Desmond Tutu TB Centre, Department of Paediatrics and Child Health, Stellenbosch University, Cape Town, South Africa, 6 Department of Infectious Diseases, Imperial College London, London, United Kingdom, 7 Centre for Bioinformatics and Computational Biology, Stellenbosch University, Cape Town, South Africa, 8 Seattle Children's Research Institute, Center for Global Infectious Disease Research, Seattle, WA, United States of America, 9 Max Planck Institute for Infection Biology, Berlin, Germany, 10 Max Planck Institute for Multidisciplinary Sciences, Göttingen, Germany, 11 Hagler Institute for Advanced Study, Texas A&M University, College Station, TX, United States of America, 12 Department of Infectious Diseases, Leiden University Medical Center, Leiden, The Netherlands, 13 South African Tuberculosis Vaccine Initiative, Institute of Infectious Disease and Molecular Medicine and Division of Immunology, Department of Pathology, University of Cape Town, Cape Town, South Africa, 14 Vaccines & Immunity Theme, Medical Research Council Unit, The Gambia, at the London School of Hygiene and Tropical Medicine, Banjul, The Gambia, 15 Catalysis Foundation for Health, San Ramon, CA, United States of America

¤a Current address: Lung Infection and Immunity Unit, Division of Pulmonology, Department of Medicine, University of Cape Town, Cape Town, South Africa
¤b Current address: Department of Experimental Medicine, University of San Francisco, San Francisco, United States of America
¶ Complete membership full author group can be found in the Acknowledgments.
* gctromp@sun.ac.za

**Data Availability Statement:** The full datasets can be obtained from the Gene Expression Omnibus (GEO; https://www.ncbi.nlm.nih.gov/geo/) as

## Abstract

*Mycobacterium tuberculosis* (*M.tb*) causes tuberculosis (TB) and remains one of the leading causes of mortality due to an infectious pathogen. Host immune responses have been implicated in driving the progression from infection to severe lung disease. We analyzed longitudinal RNA sequencing (RNAseq) data from the whole blood of 74 TB progressors whose samples were grouped into four six-month intervals preceding diagnosis (the GC6-74 study). We additionally analyzed RNAseq data from an independent cohort of 90 TB patients with positron emission tomography-computed tomography (PET-CT) scan results which were used to categorize them into groups with high and low levels of lung damage (the Catalysis TB Biomarker study). These groups were compared to non-TB controls to obtain a complete whole blood transcriptional profile for individuals spanning from early

GSE94438 (https://www.ncbi.nlm.nih.gov/geo/query/acc.cgi?acc=GSE94438) and GSE89403 (https://www.ncbi.nlm.nih.gov/geo/query/acc.cgi?acc=GSE89403). These details are also provided in the Methods. Additional data on PET-CT scores accompany the revised document as S1 Table.

**Funding:** SM, EM, GT and GW were supported by the South African Tuberculosis Bioinformatics Initiative (SATBBI), a Strategic Health Innovation Partnership grant from the South African Medical Research Council (https://www.samrc.ac.za/) and South African Department of Science and Innovation (https://www.dst.gov.za/); no grant number. STM received funding from the EDCTP2 program (Grant Number CDF1576) supported by the European Union (http://www.edctp.org/projects-2/#). GW received funding from the South African National Research Foundation (SARChI TB Biomarkers #86535) and the South African Medical Research Council (https://www.samrc.ac.za/). SHEK, TJS and GW received funding from the Bill and Melinda Gates Foundation (Grant Numbers OPP37772 & OPP1055806), (https://www.gatesfoundation.org/) GW received funding from the Bill and Melinda Gates Foundation (Grant Number OPP51919) (https://www.gatesfoundation.org/) through the Catalysis Foundation for Health (https://catalysisfoundation.org/) AGL is supported by the NRF-CSUR (Grant Number CSUR60502163639) and by the Centre for Tuberculosis Research from the South African Medical Research Council (https://www.samrc.ac.za/). JAS is supported by a Clinician Scientist Fellowship (Grant Number MR/R007942/1) jointly funded by the UK Medical Research Council (MRC; https://www.ukri.org/about-us/mrc/) and the UK Department for International Development [DFID; replaced by Foreign, Commonwealth & Development Office (FCDO); https://www.gov.uk/government/organisations/foreign-commonwealth-development-office] under the MRC/DFID Concordat agreement. The funders had no role in study design, data collection and analysis, decision to publish, or preparation of the manuscript.

**Competing interests:** The authors have declared that no competing interests exist.

stages of *M.tb* infection to TB diagnosis. The results revealed a steady increase in the number of genes that were differentially expressed in progressors at time points closer to diagnosis with 278 genes at 13–18 months, 742 at 7–12 months and 5,131 detected 1–6 months before diagnosis and 9,205 detected in TB patients. A total of 2,144 differentially expressed genes were detected when comparing TB patients with high and low levels of lung damage. There was a large overlap in the genes upregulated in progressors 1–6 months before diagnosis (86%) with those in TB patients. A comprehensive pathway analysis revealed a potent activation of neutrophil and platelet mediated defenses including neutrophil and platelet degranulation, and NET formation at both time points. These pathways were also enriched in TB patients with high levels of lung damage compared to those with low. These findings suggest that neutrophils and platelets play a critical role in TB pathogenesis, and provide details of the timing of specific effector mechanisms that may contribute to TB lung pathology.

## Introduction

*Mycobacterium tuberculosis* (*M.tb*), which causes tuberculosis (TB), remains one of the leading pathogens that is responsible for human death [1]. Although estimates suggest that 23% of the world's population are infected with *M.tb* [1], most individuals are able to eradicate or control the disease [2] and only 5–10% develop TB during their lifetime [1].

It is largely unknown why some *M.tb*-infected individuals progress to active TB, but an over-active inflammatory response is considered an important factor contributing to lung pathology [3, 4]. Rapid necrosis, associated with a delayed-type hypersensitivity reaction against accumulated *M.tb* antigens [5], or an *M.tb*-mediated autoreactive response [4], is thought to cause an inflammatory response that drives lung extracellular matrix (ECM) destruction and cavity formation. This allows *M.tb* in the lung interstitium to access the airways and be transmitted [6].

Neutrophils are strongly activated in response to *M.tb* infection [7, 8], although numerous studies have shown they are ineffective at killing or controlling *M.tb* replication (for review see [9]). Rather, the activation and infiltration of neutrophils at late stages of infection is associated with TB pathogenesis [9–13]. Indeed, neutrophils are the predominant immune cell type present in lung lesions and cavities of pulmonary TB patients and are associated with lung ECM destruction and cavity formation [12, 14]. Neutrophils contain a diverse array of preformed proteases in their granules including neutrophil collagenase and matrix metallopeptidase 8 (MMP8), which digest ECM in human lung [12] and MMP8 is the most prevalent MMP present in the sputum of TB patients [15]. *M.tb*-induced MMP8 secretion is also associated with the secretion of neutrophil extracellular traps (NETs); a process by which neutrophils release their antimicrobial granule proteases, DNA and histones extracellularly in a type of programmed cell death named NETosis [16]. Collectively, these studies strongly implicate neutrophils in late stages of TB lung pathology including ECM destruction and cavity formation.

Whole blood transcriptomic studies have advanced our knowledge of the host response to many diseases, including TB [11, 17–19]. Previous transcriptomics studies identified a number of immune processes activated in TB patients including interferon (IFN)-signaling [11, 17, 20], myeloid cell inflammation [21], and the inflammasome and proinflammatory pathways [17]. Two large cohort studies that monitored individuals at high risk for developing TB for up

to two years before diagnosis generated whole blood transcriptional profiles for TB progressors with the aim to develop predictive transcript-based risk signatures [22, 23]. Subsequently, data from one of these studies, the South African adolescent cohort (aged 12–18 years) [23], was analyzed to identify biological pathways and processes that are active during TB progression [24, 25]. These studies found that type I IFN signaling and the complement cascade were the main pathways activated at early stages of TB progression as observed in TB patients [24, 25] and changes in functionally uncharacterized neutrophil and platelet gene modules occurred at times closer to TB diagnosis [25]. Here we extended the prior analyses by jointly analyzing two data sets and dissecting the neutrophil processes.

We performed a differential expression (DE) and comprehensive pathway analysis of whole-blood RNA sequencing (RNAseq) datasets obtained from the Gene Expression Omnibus (GEO) [26]. The first set, GEO series GSE94438 (GC6-74 study), was generated from individuals for a period of up to two years prior to TB diagnosis [22]. The second data set, GEO series GSE89403 (Catalysis TB Biomarker Study, hereafter the Catalysis study), compared TB patients to healthy controls. Collectively, the data sets provide a full spectrum of host transcriptional responses spanning from early infection stages to TB diagnosis. Our analysis focused on identifying host responses that may contribute to the development of TB.

## Results

### Study groups and data quality

We performed a DE and comprehensive pathway analysis of whole-blood RNAseq data sets obtained from the GEO.

The first data set included RNAseq data for TB progressors (GEO:GSE94438) and was generated as part of the Grand Challenges 6–74 (GC6-74) program. The program was a longitudinal study of household contacts of newly diagnosed, sputum smear-positive TB cases in a high TB-prevalence settings and included samples from four different African populations. A detailed description of the study groups has been published previously [22, 23, 27, 28]. Contacts diagnosed with TB within 3 months of recruitment were considered to have prior disease and were excluded from the study, while those developing disease after this point were considered to have true incident TB and were included as cases. Due to the finite follow-up some of the subjects classified as non-progressor might have had pre-clinical TB. Such subjects in the "control" group would dilute the disease signals, possibly obscuring some insights, but would not alter the fundamental findings.

Since the aim of the present study was to identify genes that are DE at different stages of TB progression, the progressor samples were labelled according to the time the sample was collected before their TB diagnosis. As an example, if a participant was diagnosed with TB 8 months after enrollment and they had samples collected at enrollment (time 0) and 6 months, these samples would be classified as 8 and 2 months before diagnosis, respectively. The labelled samples were subsequently placed into groups that corresponded to 19–24, 13–18, 7–12 and 1–6 months before TB diagnosis (longitudinal data for subjects who were diagnosed with TB were aligned to the time of diagnosis and specimens taken before diagnosis were binned into the aforementioned groups). The number of samples in each study group, and mean time before TB diagnosis for each time point are listed in Table 1. For the GC6-74 study, all the TB progressor groups were compared to the same non-TB control samples that were collected at recruitment time. This ensured that any differences between the progressor groups were a result of changes in their respective expression rather than changes in the baseline group.

The second data set used included a subset of RNAseq data from the South African Catalysis study (GEO: GSE89403). Here, we only used RNAseq data from TB patients at diagnosis and the

**Table 1. Study groups.**

| | | GC6-74 Study[a] | | | | | Catalysis Study[b] | |
|---|---|---|---|---|---|---|---|---|
| | | Time[c] before TB diagnosis in progressors | | | | | | |
| | | **19–24** | **13–18** | **7–12** | **1–6** | **Control[d]** | **TB** | **Control[e]** |
| N | | | | | | | | |
| | Total | 11 | 19 | 18 | 47 | 198 | 90 | 21 |
| | Male | 3 | 7 | 8 | 20 | 81 | 55 | 8 |
| Age | | | | | | | | |
| | Mean (y) | 24.2 | 25.5 | 27.9 | 29.0 | 27.5 | 34.3 | 33.4 |
| | SD | 9.4 | 11.3 | 11.3 | 12.0 | 13.0 | 11.2 | 11.5 |
| Time to diagnosis | | | | | | | | |
| | Mean | 21.5 | 16.1 | 9.2 | 4.2 | NA[f] | NA | NA |
| | SD | 0.6 | 0.4 | 0.4 | 0.2 | | | |
| | 95% CI | 20.4–22.6 | 15.3–16.8 | 8.3–10.1 | 3.8–4.6 | | | |

[a]GEO series GSE94438

[b]GEO series GSE89403

[c]Time in months before diagnosis

[d]GC6-74 Controls: household contacts who did not develop TB

[e]Catalysis Controls: Other lung disease subjects

[f]NA, Not applicable

non-TB controls [27, 28]. This data set also contained positron emission tomography-computed tomography (PET-CT) data for the TB patients which we used to categorise the patients (see Methods for details) into groups with high (high PET scores) and low (low PET scores) levels of lung damage. The primary analysis for the Catalysis study data set compared all TB patients to controls. We additionally performed a number of PET-score based sub-analyses where patients with high and low PET scores were individually compared to controls, and patients with high and low PET scores were directly compared to identify differences between them.

The FastQC analysis revealed that the read quality was good with no adapter contamination, therefore, no trimming was performed. Mapping statistics revealed that for the GC6-74 cohort, a mean of 47.5 million reads [(47.1, 47.9: 95% CI; 85.9% (95% CI: 85.7, 86.1)] mapped uniquely to the human genome with 87.8% (95% CI: 87.7, 87.9) of these mapping to genes. For the Catalysis cohort, a mean of 40.1 million reads [(39.4, 40.8: 95% CI; 88.6% (88.3, 88.9: 95% CI)] mapped uniquely to the human genome with 88.8% (95% CI: 88.7, 88.9) mapping to genes.

## Differential expression analysis

The edgeR DE analysis identified several significantly perturbed genes across the different time points (Table 2). For the progressors in the GC6-74 study [23], there was a steady increase in the number of DE genes detected at time points more proximal to TB diagnosis with only five genes at 19–24 months, 278 at 13–18 months, 742 at 7–12 months and 5,131 at 1–6 months before diagnosis. A total of 9,205 genes were DE in TB patients compared to controls (Catalysis study). In the Catalysis sub-group analysis, in comparison to controls, a similar number of genes were significantly DE in patients with high (9,211) and low (8,496) PET-scores (Table 2). Notably, a total of 2,144 significantly DE genes were identified when directly comparing patients with high and low PET-scores confirming differences in transcriptional signatures between TB patients with high and low levels of lung damage.

**Table 2. Number of significantly differentially expressed (DE) genes (FDR<0.05) at each time point.**

| Test group | GC6-74 Study[a] | | | | Catalysis Study | | | |
|---|---|---|---|---|---|---|---|---|
| | 19–24[b] | 13–18[b] | 7–12[b] | 1–6[b] | TB[c] | PET high[c] | PET low[c] | PET high[d] |
| Case N | 11 | 19 | 18 | 47 | 90 | 24[e] | 65[e] | NA |
| DE Gene Count | | | | | | | | |
| Up | 5 | 247 | 492 | 2,791 | 4,200 | 4,284 | 4,031 | 995 |
| Down | 0 | 31 | 250 | 2,340 | 5,005 | 4,927 | 4,465 | 1,149 |
| Total | 5 | 278 | 742 | 5,131 | 9,205 | 9,211 | 8,496 | 2,144 |

[a] Reference group: GC6-74 non-TB controls

[b] Months before diagnosis

[c] Reference group: Catalysis non-TB controls

[d] Reference group: Catalysis PET-low

[e] One TB patient in the Catalysis Study group did not have PET-CT data

The DE results for all genes and time points are presented in **S2 Table**. There was substantial overlap in the DE genes with 92% (445/492) of the genes upregulated 7–12 months before diagnosis also being upregulated in the samples taken 1–6 months prior to TB diagnosis. Similarly, 85% (2,365/2,791) of the genes upregulated 1–6 months before diagnosis in GC6-74 study subjects were also upregulated in Catalysis study TB patients (Fig 1A). The high level of overlap in genes upregulated 1–6 months before diagnosis with those upregulated in TB patients illustrates substantial homogeneity in the host response between the two cohorts and provides support for the validity in their comparisons.

In the Catalysis study PET categorized sub-analysis, the TB high PET score group vs controls shared 91% (3,818/4,200) while the TB low PET score group vs controls shared 94% (3,957/4,200) of the upregulated genes identified in the complete analysis (Fig 1B). Of the 995

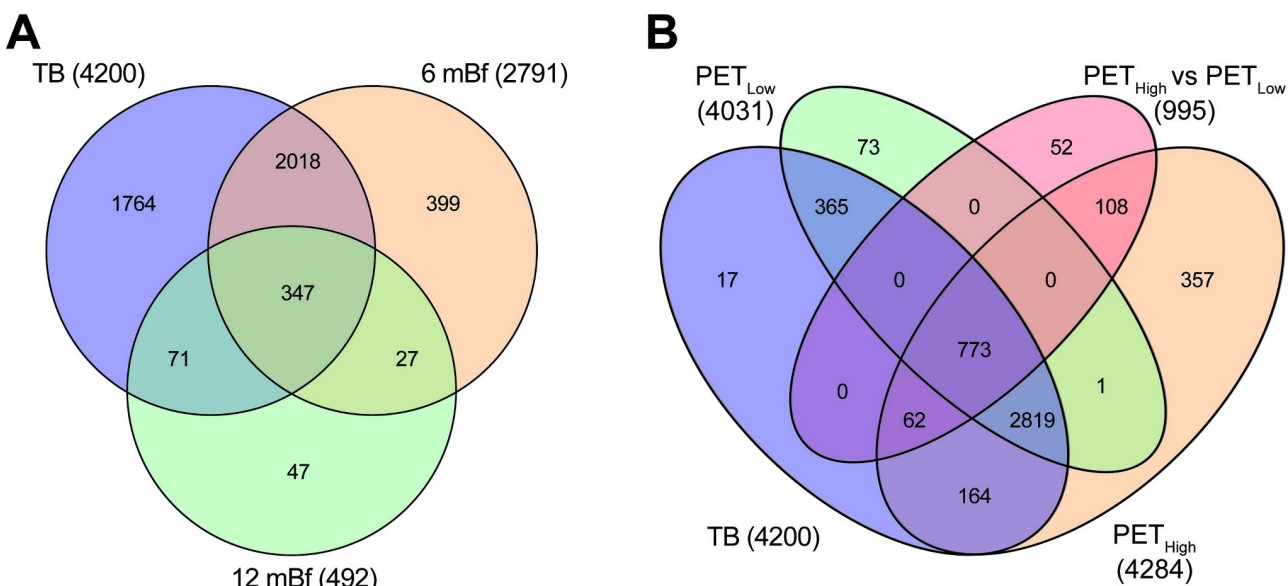

**Fig 1. Venn diagram illustrating the overlap in significantly upregulated genes.** A) Comparison of upregulated genes detected 7–12 and 1–6 months before diagnosis and in TB patients. B) Comparison of upregulated genes detected in TB patients with high and low PET scores compared separately to controls and high and low PET scores compared to each other. The numbers in parenthesis indicate the total number of upregulated genes for the specific comparison.

genes that were upregulated when directly comparing the TB high PET score group to the TB low PET score group, 95% (943/995) were upregulated when comparing the TB high PET score group to controls and 78% (773/995) were upregulated when comparing the TB low PET score group to controls. Thus, 773 genes that were upregulated when comparing the TB low PET score group to controls, were further significantly upregulated in TB patients with high PET scores compared to those with low PET scores.

Since the increased sample size at time points more proximal to TB diagnosis could be responsible for the increased number of DE genes observed, we performed subsampling to confirm that true biological differences were responsible for the increased number of genes. The subsampling of the 1–6 months before diagnosis group revealed that on average 1,454 genes were upregulated across 1,000 iterations of 19 samples compared to 2,791 upregulated genes detected with a sample size of 47 (S1 Fig). For the TB group, a mean of 3,510 genes were upregulated across 1,000 iterations of 19 samples compared to 4,200 upregulated genes detected with a sample size of 90 (S2 Fig). The higher mean number of upregulated genes detected when subsampling both groups compared to the 13–18 (249; n = 19) and 7–12 (492; n = 18) months before diagnosis groups (Table 2) indicated that the increased number of DE genes observed was predominately a consequence of the time point. This is also supported by almost double the number of genes detected in the 7–12 compared to the 13–18 months before diagnosis groups which had similar sample sizes. The results also confirmed that larger sample numbers improve statistical power and facilitate the detection of additional DE genes.

### Gene Ontology (GO) and KEGG pathway analysis

Since only five genes were DE 19–24 months before diagnosis (S2 Table), the pathway analysis was performed on the 13–18, 7–12 and 1–6 months before diagnosis time points and on TB patients. The use of the weighted algorithm (weight01) in topGO identified the most specific and thus informative GO terms including several not previously reported during TB progression. In addition, only pathways that had a minimum of 10 DE genes were considered. In summary, the pathway analysis of the upregulated genes identified several defense-related pathways that were enriched early (13–18 months before diagnosis), remained elevated and intensified, based on increased numbers of annotated genes, at time points more proximal to TB diagnosis. Several additional processes that were specifically related to neutrophil and platelet defense responses were first enriched 1–6 months before diagnosis and remained enriched in TB patients.

A list of the GO and KEGG pathways significantly perturbed at different stages of TB progression is provided in S2 and S3 Tables. Here, in the main text, we focus on the induction of neutrophil-mediated defenses and related pathways and processes since we consider these the most likely drivers of disease progression.

### Early (13–18 and 7–12 months before) pathway activation

The most enriched GO terms detected 13–18 and 7–12 months before diagnosis were associated with IFN signaling including "type I interferon signaling pathway" and "interferon-γ-mediated signaling pathway" (Fig 2A).

### One to six months before diagnosis

A total of 5,131 genes were significantly DE 1–6 months before TB diagnosis with respect to household controls with 2,791 (54%) up-regulated (Table 2). The marked increase in the number of DE genes was notably accompanied by a corresponding enrichment in several effector pathways that were not enriched at earlier time points documenting a strong and distinct

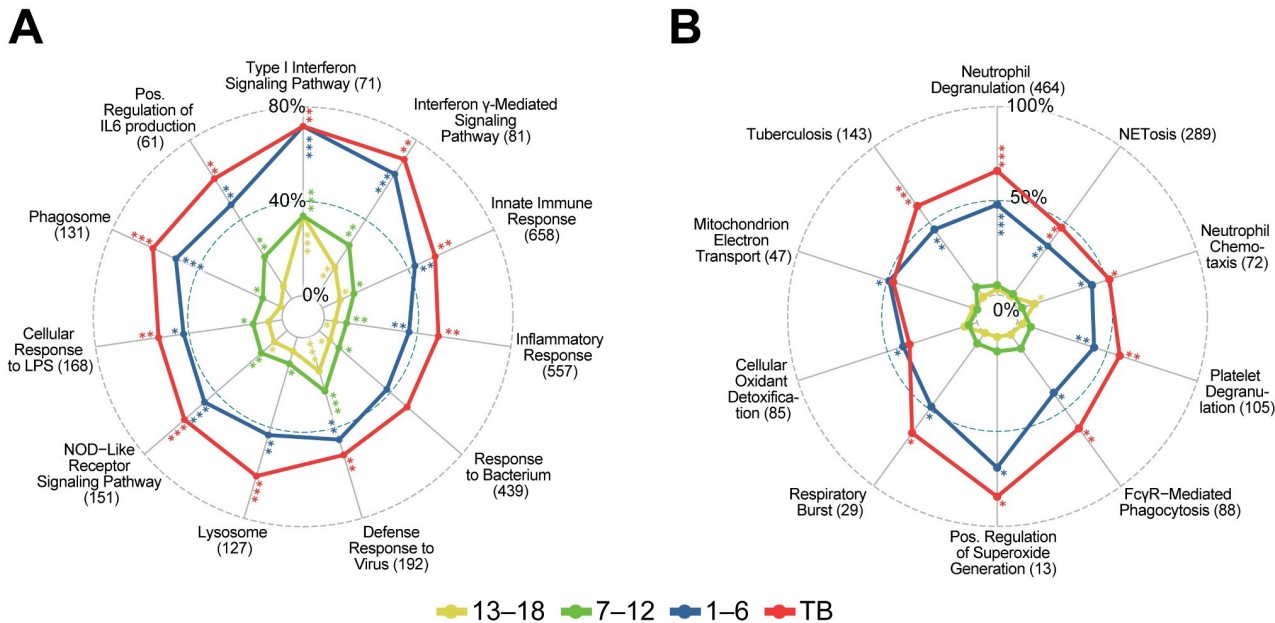

**Fig 2.** GO annotations and KEGG pathways first enriched at (A) 13–18 and 7–12 months before diagnosis and (B) 1–6 months before diagnosis. Radar plots illustrating proportions of DE genes in the gene sets at the indicated time points for selected pathways broadly related to neutrophil function. All four time points [13–18, 7–12, and 1–6 months before, and at diagnosis (TB)] are shown in each plot to demonstrate that annotations detected early persist and generally have more DE genes at later stages, although significance may decrease. The numbers in parenthesis represent the total numbers of genes in the annotation gene set. Asterisks indicate nominal significance: ***, $< 5e^{-15}$; **, $< 5e^{-07}$; *, $< 5e^{-04}$.

activation of the host immune response at this time point. This response was dominated by a strong enrichment in up-regulated genes that function in neutrophil-mediated immunity with "neutrophil degranulation" [$p<10^{-30}$, 222/464 (48%) annotated genes] being the most significant and specific GO biological process (BP) term detected (Fig 2B). Other enriched pathways related to neutrophil function included "neutrophil chemotaxis", "Fc-gamma receptor signaling pathway involved in phagocytosis" and "platelet degranulation". Several terms related to the production of reactive oxygen species (ROS) were also enriched including "positive regulation of superoxide anion generation", "respiratory burst", "mitochondrial electron transport" and "cellular oxidant detoxification". The KEGG "NET formation" and "tuberculosis" disease pathways were also enriched at this time point (Fig 2B). An illustration of the KEGG "NET formation" pathway is presented in Fig 3A and 3B with fold-changes (log2) for significantly upregulated genes displayed for the different time points.

Annotated neutrophil degranulation genes upregulated 1–6 months before diagnosis included neutrophil *integrin subunit beta 2* (*ITGB2/LFA1*), *integrin subunit alpha M* (*ITGAM*) and *integrin subunit alpha X* (*ITGAX*) that form the transmembrane heterodimer integrin receptors ITGAM/ITGB2 (mac-1) and ITGAX/ITGB2 that facilitate the transmigration of neutrophils to infection sites [30] (Fig 4). The neutrophil granule transmembrane chemotactic *formyl peptide receptor* (*FPR*) *1* and *2* that bind formyl-peptides derived from bacteria and damaged host molecules and the *CXCR1* and *2* receptors that bind the neutrophil chemotactic and activation factors IL8 and CXCL7 (neutrophil-activating peptide-2, NAP2) were also induced. Ligation of the G protein-coupled receptors FPR1 and 2, and CXCR1 and 2 induces calcium signaling which regulates neutrophil migration, ROS synthesis and primes neutrophils for further activation and the release of subsequent granules [31]. The genes encoding the antimicrobial myeloid-related proteins (MRP), namely *S100A8*, *S100A9*, *S100A11*, and *S100A12*, were also upregulated [16, 32].

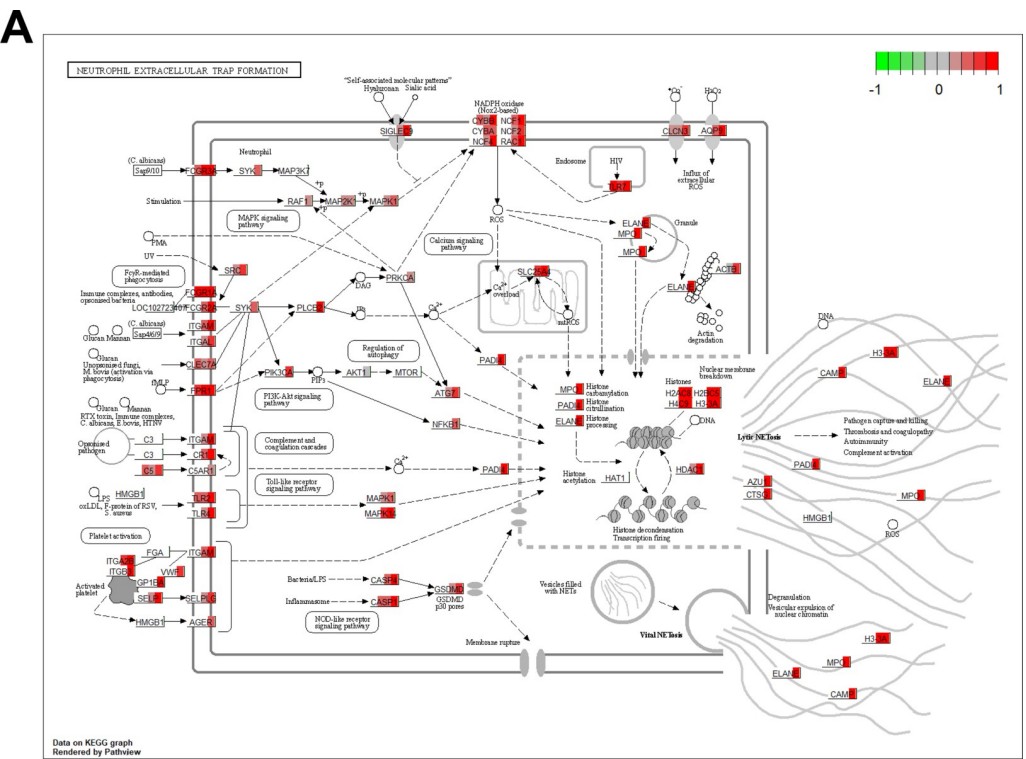

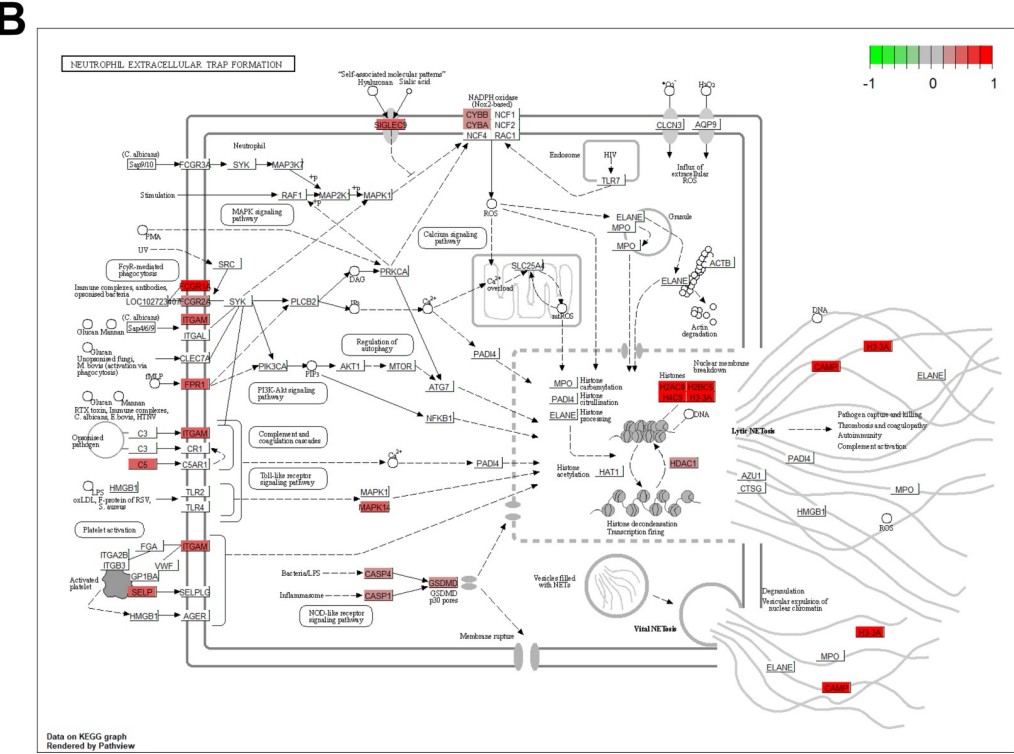

**Fig 3. KEGG NET formation pathway illustrating the fold changes (log2) for significantly upregulated genes at different time points.** A) Gene expression as fold-change is indicated by a color block in the background rectangle or each gene corresponding to the contrasts, 13–18, 7–12, 1–6 months before diagnosis and TB compared to their respective controls (from left to right). B) As above but illustrating the significantly increased expression of genes in TB patients with high PET-scores compared to those with low PET-scores. Figures rendered by the R Bioconductor pathview package [29], on finite resolution raster images from KEGG.

| Six Months Before | TB | PET$_{High}$ | PET$_{Low}$ | PET$_{High}$ vs PET$_{Low}$ | |
|---|---|---|---|---|---|
| 0.42 | 0.58 | 0.69 | 0.49 | 0.21 | CYBA |
| 0.34 | 0.88 | 1.06 | 0.74 | 0.32 | CYBB |
| 0.56 | 1.13 | 1.25 | 1.04 | | FCGR3B |
| 0.52 | 0.97 | 1.17 | 0.84 | 0.33 | FCGR2A |
| 0.49 | 1.05 | 1.3 | 0.92 | | FCAR |
| 0.23 | 0.69 | 0.72 | 0.62 | | ITGB2/mac−1/LFA1 |
| 0.3 | 1 | 1.25 | 0.85 | 0.4 | ITGAM/mac−1 |
| 0.22 | 0.72 | 0.93 | 0.59 | 0.34 | ITGAX |
| 0.55 | 0.97 | 1.17 | 0.86 | | FPR1 |
| 0.41 | 0.9 | 1.17 | 0.74 | 0.43 | FPR2 |
| 0.49 | 0.83 | 1.02 | 0.72 | | TLR2 |
| 0.45 | 1.19 | 1.47 | 1.05 | | CR1 |
| 0.6 | 1.26 | 1.57 | 1.1 | | C3AR1 |
| 0.57 | 1.08 | 1.28 | 0.96 | 0.32 | SELL |
| 0.5 | 0.86 | 0.94 | 0.8 | | CXCR1 |
| 0.28 | 0.62 | 0.61 | 0.57 | | CXCR2 |
| 2.24 | 3.7 | 4.43 | 3.49 | | CD177 |
| 0.47 | 1.45 | 2.04 | 1.05 | 0.99 | PPBP/NAP−2/CXCL7 |
| 0.22 | 0.39 | 0.34 | 0.38 | | PTAFR |
| 0.32 | 1.04 | 1.28 | 0.91 | 0.37 | LTA4H |
| 0.25 | 0.8 | 0.9 | 0.7 | | ALOX5 |
| 0.57 | 1.03 | 1.16 | 0.94 | | S100A11 |
| 0.92 | 1.62 | 2.33 | 1.47 | 0.86 | S100A12 |
| 0.9 | 1.53 | 2.06 | 1.33 | 0.73 | S100A9 |
| 0.91 | 1.42 | 2.09 | 1.26 | 0.83 | S100A8 |
| 0.46 | 0.72 | 0.83 | 0.64 | | LYZ |
| 0.46 | 1.1 | 1 | 1.01 | | PADI2 |
| | 2.43 | 3.5 | 1.92 | 1.59 | MMP8 |
| | 1.36 | 1.53 | 1.21 | | MMP9 |
| | 0.85 | 0.98 | 0.73 | | MMP25 |
| | 0.8 | 0.97 | 0.72 | | MME |
| | 1.53 | 1.71 | 1.4 | | MPO |
| | 2.01 | 2.28 | 1.84 | | ELANE |
| | 1.53 | 1.54 | 1.43 | | AZU1 |
| | 2 | 2.59 | 1.62 | | DEFA4 |
| | 2.11 | 2.31 | 1.96 | | PRTN3 |
| | 2 | 2.62 | 1.64 | | LTF |
| | 1.69 | 2.12 | 1.36 | | BPI |
| | 1.58 | 1.68 | 1.47 | | CTSG |
| | 1.27 | 2.01 | 0.96 | 1.04 | CAMP/LL37 |
| | 1.95 | 2.59 | 1.56 | 1.03 | LCN2/NGAL |

1    3
FC (log$_2$)

**Fig 4. Heatmap of selected annotated neutrophil degranulation genes significantly upregulated 1–6 months before diagnosis and in TB patients.** The fold changes presented at 1–6 months before diagnosis (6mBf) are in contrast to the non-TB time zero baseline group from the GC6-74 study. For the TB comparison, the 21 non-TB individuals from the Catalysis study served as controls (see Table 1). Fold-changes for Catalysis TB-patients, categorized with high and low PET scores, were also compared individually to controls and directly to each other to identify differences between the categorized groups.

Additional upregulated genes annotated to function in neutrophil degranulation include those encoding components of the superoxide generating NADPH oxidase 2 (*NOX2*). This included the transmembrane catalytic [cytochrome b-245 -alpha (*CYBA*) and -beta (*CYBB*)] and cytosolic regulatory subunits [neutrophil cytosolic factor (*NCF*)*1*, *2* and *4* and the small G-protein, *RAC2* [33, 34]. The induction of genes encoding components of NOX2 along with the enrichment of pathways involved in superoxide generation and the respiratory burst, is consistent with the enrichment of FcγR mediated phagocytosis which activates NOX2 assembly [34, 35]. NOX2 generated ROS also induce the synthesis of NETs [36, 37].

The co-induction of "platelet degranulation" with "neutrophil degranulation", first observed 1–6 months before diagnosis, is consistent with the physical interactions and costimulatory roles platelets and neutrophils share during immune responses [38, 39].

## TB versus healthy controls

A total of 9,205 genes were DE in individuals with TB in contrast to the non-TB time zero baseline group from the GC6-74 study, of which 4,200 (46%) were up-regulated (Table 2).

Similar to 1–6 months before diagnosis, there was a strong enrichment in pathways related to neutrophil-mediated immunity with "neutrophil degranulation" [p<1x10$^{-30}$, 305/464 (66%) annotated genes] being the most significant and specific GO BP term detected (Fig 2A). Approximately 95% (210/222) of the neutrophil degranulation (Fig 5A) and 97% (60/62) of NET formation (Fig 5B). annotated genes that were upregulated 1–6 months before diagnosis were also upregulated in TB patients. All the other enriched neutrophil related functions observed 1–6 months before diagnosis were also enriched in TB patients.

Several additional terms related to neutrophil function were enriched in TB patients, but not 1–6 months before diagnosis. This included the terms "leukocyte transendothelial migration", "neutrophil extravasation", "extracellular matrix disassembly", "collagen catabolic process", "antibacterial humoral response", "antimicrobial humoral immune response mediated by antimicrobial peptide", "platelet activation" and "platelet aggregation" (Fig 6).

It is interesting to note that the expression of several genes that encode neutrophil granule antimicrobial peptides were only elevated in TB patients and not progressors (Fig 4). These included myeloperoxidase (*MPO*), neutrophil elastase (*ELANE*), cathepsin G (*CTSG*), cathelicidin antimicrobial peptide (*CAMP/LL37*), lactotransferrin (*LTF*), defensin alpha 4 (*DEFA4*), azurocidin 1 *(AZU1)*, bactericidal/permeability-increasing protein (*BPI*), *MMP8*, *MMP9*, and neutrophil gelatinase-associated lipocalin (*NGAL*). A number of these genes are also annotated to function in other enriched pathways including NET formation and antimicrobial and antibacterial humoral defense responses.

### Catalysis PET-CT sub-analysis

Unsurprisingly, given the high level of overlap between the genes, the results of the pathway analysis were very similar to those observed when comparing the complete TB group to controls, with "neutrophil degranulation", "platelet degranulation" and "NET formation" being significantly enriched in both the TB-high-PET score group and the TB-low-PET score group

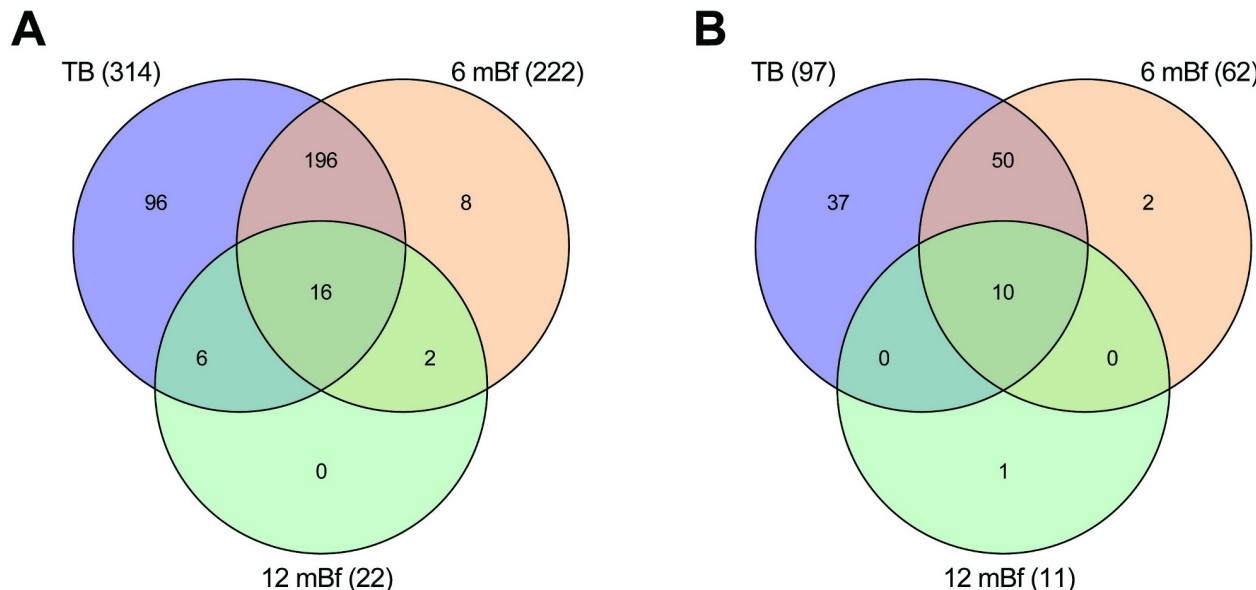

**Fig 5.** Venn diagram of the overlap in significantly upregulated genes that are annotated to function in: A) neutrophil degranulation, and B) NET formation. Differentially expressed genes at 7–12 and 1–6 months before diagnosis and in TB patients in contrast to the non-TB time zero baseline group from the GC6-74 study. The numbers in parenthesis indicate the total numbers of upregulated genes at the time point.

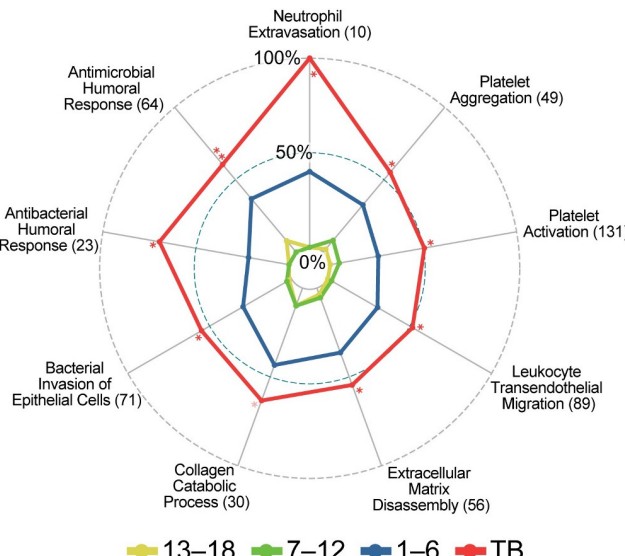

**Fig 6. Pathways first enriched in TB patients.** Radar plot with symbols as in Fig 2. The full name for "Antimicrobial Humoral Immune Response . . ." is "Antimicrobial humoral immune response mediated by antimicrobial peptide (GO:0061844)".

vs controls comparisons. Strikingly, these pathway were also enriched when directly comparing the TB-high-PET score group to the TB-low-PET score group with neutrophil degranulation [p = $8.1^{-19}$, 86/464 (19%)], platelet degranulation [p = $4.5^{-6}$, 21/105 (20%)] and NET formation [p = $5.8^{-12}$ 34/142 (24%), Table 3)], revealing that TB patients with increased lung damage have an enrichment in upregulated genes that function in these pathways. Annotated neutrophil degranulation genes that were significantly upregulated in TB patients with high PET scores compared to those with low PET scores included *MMP8*, *S100A12*, *S100A8*, *S100A9*, *CAMP* and *PPBP* (Fig 4).

## Discussion

Our results revealed an early induction of IFN-related signaling at 18 months before diagnosis. A strong induction of neutrophil and platelet degranulation and NETosis related genes was detected 6 months before TB diagnosis and persisted in TB patients supporting a pathogenic role of these responses in disease development. Although these processes have previously been detected in patients diagnosed with active TB, here we document that they occur well before diagnosis, indicating that they may be critical in mediating lung tissue destruction and progression from infection with M.tb to active TB.

**Table 3. Pathways significantly enriched with upregulated genes when comparing TB patients with high PET-scores to TB patients with low PET-scores.**

| Pathway | Total annotated genes | Number Up | % Up | P-value | Rank |
|---|---|---|---|---|---|
| Neutrophil degranulation[a] | 464 | 86 | 18.5 | $1.2^{-18c}$ | 1 |
| Platelet degranulation[a] | 105 | 21 | 20 | $4.5^{-06c}$ | 8 |
| NET formation[b] | 142 | 34 | 23.9 | $5.8^{-12}$ | 2 |

[a]GO biological process

[b]KEGG pathway

[c]weight01.algorithm p-value

DE and pathway enrichment analysis revealed that an induction of the host defense response, that is dominated by type I and II IFN signaling, was first detectable in whole blood of TB progressors as early as 13–18 months before diagnosis. A progressive induction in the intensity of the IFN response was observed as time points approached TB diagnosis with a distinct induction of neutrophil-mediated defense observed 1–6 months before diagnosis that persisted and intensified in patients diagnosed with TB. Given that we examined transcriptional responses in whole blood, changes observed are likely to result from signaling molecules released from infected cells, most likely in the lungs, rather than through direct contact with the pathogen. The signaling is likely to prime circulating leukocytes for activation in preparation for their recruitment to infection sites.

A unique and striking finding was the strong induction of specific neutrophil-mediated defenses, including neutrophil degranulation, and NET formation, that are first observed 1–6 months before diagnosis (Fig 2B), maintained in TB patients and significantly enriched in TB patients with high levels of lung damage compared to those with low level damage assessed by PET-CT (S4 Table). The induction of neutrophil-mediated defenses is consistent with literature that associates neutrophil activation and infiltration with TB as well as pulmonary destruction [9, 11, 13]. Neutrophils have been reported to be the predominant immune cell type present in the sputum and at the site of infection in the lung [40], and are also the main cell type infected with *M.tb* in sputum and lung cavities [14]. In addition, neutrophil markers are associated with necrotic areas in granulomas [41] and excessive neutrophil infiltration is associated with the softening of caseous lesions in the lung [5].

Although the neutrophil degranulation and NET formation processes are both enriched 1–6 months before diagnosis and in TB patients, the additional increase in the expression level and induction of additional genes that function in these pathways in TB patients suggests a sequential induction of neutrophil activation during the late stages of disease progression. Genes induced 1–6 months before diagnosis encode neutrophil membrane integrins and chemotactic receptors that mediate their adhesion and transmigration to infection sites as well as components of NOX2 [42]. These genes are generally involved in the priming of neutrophils in preparation for full activation at infection sites. In TB patients, there is a distinct induction of genes that encode neutrophil granule proteins that function in ECM degradation including *MMP8* and *MMP9* and those that encode numerous neutrophil azurophilic granule proteins including *MPO*, *ELANE*, *DEFA4* and *BPI*. The MPO, ELANE and PADI4 peptides play a critical role in NET formation, driving chromatin decondensation, cell rupture and the extracellular release of DNA [43–45]. The significantly higher induction of genes in patients with high-PET scores compared to those with low-PET scores, including *MMP8*, links their increased expression to increased lung damage.

The induction of genes that encode neutrophil granule proteins, including bactericidal granule enzymes, is intriguing since expression of these genes is typically high in neutrophils during maturation when granule proteins are synthesized, and declines in mature cells [46, 47]. A highly similar set of genes including *AZU1*, *CAMP*, *CTSG*, *DEFA4*, *ELANE*, *LTF, and MPO* is activated and forms part of a co-expression module in isolated low-density granulocytes (LDG) from systemic lupus erythematosus (SLE) patients and it is thought that LDGs are immature neutrophils that have been released into the circulation during granulopoiesis [45, 48]. Elevated LDG levels are correlated with disease severity in TB patients, but *in vitro* studies indicated LDGs were generated from normal neutrophils after degranulation or NET formation [49, 50]. These studies, however, did not investigate whether the expression of genes encoding the granule proteins were elevated in these cells. Given our analysis is on whole blood which includes multiple cell types, it is possible that the observed elevated expression of neutrophil related genes is a result of an increase in the proportion of neutrophils in the blood. This, however,

could only partially account for the observed increases since the expression of a number of these genes including *ELANE*, *DEFA4*, *BPI* and *LTF* (Fig 4) are elevated > 4-fold (log2 FC > 2). Since neutrophils already constitute 50% to 70% of all circulating leukocytes, an increase in their abundance could not possibly account for the increased transcription observed. Nevertheless, the increased expression of neutrophil granule genes observed in this study may contribute to tissue destruction and TB pathogenesis since pro-inflammatory LDGs have an enhanced capacity to secrete NETs and granule peptides in TB and SLE patients [45, 50].

While *M.tb* has been shown to stimulate NET synthesis [51], NETs have a limited ability to kill *M.tb* [52] and it has been suggested they rather provide a platform for extracellular *M.tb* replication that facilitates pulmonary lesion growth and drives the transition to infectious TB [9, 53, 54]. In TB patients' plasma NET, MPO and ELANE levels are correlated with TB severity [55], while elevated serum levels of citrunillated histone H3, a NET biomarker, are associated with lung cavitation and poor treatment outcome [56]. A number of NET components, including dsDNA, mitochondrial DNA, and granule proteinases function as immune-stimulatory molecules when released extracellularly [45] and have been identified as important drivers of immune-pathogenesis in both infectious and non-infectious human diseases [57].

[61, 62] Given the early and sustained level of type I IFN signaling observed during TB progression, it is interesting to note that in TB susceptible mice, type I IFN signaling has been shown to induce NETosis through activation of interferon α and β receptor subunit 1, which is associated with enhanced mycobacterial growth at infection sites and enhanced TB pathogenesis [58]. The same study identified NETs in nectrotic lung lesions of TB patients that responded poorly to treatment. Further, serum from patients with autoimmune disorders that have elevated levels of type I IFNs, as well as exogenous IFN-α, has been shown to stimulate neutrophil NET production in vitro suggesting that type I IFN prime neutrophils for NET production [59–61]. In turn, self-DNA and antimicrobial peptides released with NETs, have been reported to induce the chronic activation of plasmacytoid dendritic cells and secretion of type I IFNs in SLE patients creating a positive feedback loop that prolongs the inflammatory response [62]. The upregulated type I IFN signaling observed in this study that precedes neutrophil activation and NETosis, along with the above-mentioned studies, is consistent with type I IFN signaling diving neutrophil activation and NETosis during TB progression and enhancing TB pathogenesis.

The co-induction of platelet and neutrophil-mediated defense responses starting as early as 13–18 months before diagnosis is consistent with their established dependent roles during immune responses [38, 39, 63]. The activation of platelet and neutrophil degranulation, including the induction of platelet and neutrophil derived granule chemokines and membrane proteins that mediate their co-migration and physical adhesion including *P-selectin* (*SELP*) and *CXCL7* (*neutrophil-activating peptide-2 NAP2*) from platelets and *selectin P ligand* (*SELPLG*), *CXCR1* and *2*, and *ITGB2/LFA1* from neutrophils is consistent with studies that document their physical interactions during defense responses [38, 63]. Platelet-neutrophil adhesion induces intracellular signaling cascades that activate many neutrophil antimicrobial functions observed in this study including ROS production, phagocytosis and NETosis [39, 64]. Indeed, the neutrophil membrane integrin ITGB2/LFA1 is required to mediate neutrophil-platelet adhesion that drives NET release in human sepsis [64]. It has been suggested that platelets function as a barometer, stimulating NET synthesis when bacterial levels exceed the neutrophils' capacity to control infection through alternative mechanisms [64, 65].

Platelet-neutrophil complexes are implicated in the pathogenesis of pulmonary inflammation and acute lung injury [39]. In TB patients, platelet numbers and activity are increased [66, 67] and the concentration of numerous platelet-derived mediators including P-selectin, RANTES and PDGF was increased and correlated with levels of tissue-degrading MMPs 1, 7,

8, and 9, in bronchoalveolar lavage samples [68]. The co-induction of platelet and neutrophil functions that are first observed 1–6 months before diagnosis and further activated in TB patients suggests a pathological role for these interactions in the late stages of TB development. This is further supported by the increased induction of these pathways in TB patients with high PET scores compared to those with low PET scores.

The current findings confirm and elaborate the findings of a previous study by Scriba et al. [25] that identified the induction of uncharacterized neutrophil and platelet gene modules 6 months before TB diagnosis. Here, we identified specific processes that are mediated by these cells, including neutrophil and platelet degranulation as well as NET formation that are activated 6 months before TB diagnosis. Our analyses additionally discovered that these processes are further activated in individuals with TB and at an elevated level in TB patients with increased lung damage. Collectively, these results support that the activation of these pathways is linked to TB progression and increased lung damage in TB patients.

## Conclusion

The distinct co-induction of neutrophil and platelet degranulation as early as 13–18 months before diagnosis, NET formation 1–6 months before diagnosis, as well as their further activation in TB patients is consistent with these processes playing a critical role in the late stages of disease progression. This is further supported by the enrichment in upregulated genes that function in these pathways in TB patients with increased levels of lung damage. Platelet-neutrophil interactions are required for mediating their chemotaxis and recruitment to infection sites and for NET synthesis [69]. NETs are associated with TB pathogenesis and are thought to provide an extracellular platform for *M.tb* growth [53] while neutrophil granule enzymes degrade the ECM of the lung [12]. Collectively, these responses can lead to rapid lesion growth and tissue destruction that allows *M.tb* to disseminate into the airways [53]. The detectable activation of these specific processes in whole blood around 6 months before TB diagnosis makes them promising candidates for targeted therapeutic interventions that may limit lung damage and prevent progression to active TB.

## Methods

### Study outline and data sources

The whole blood RNAseq data analyzed was obtained from two independent data sets.

The data for TB progressors were generated as part of the the Bill and Melinda Gates Foundation GC6-74 program that was a longitudinal study of household members of newly diagnosed TB cases which was conducted across four African sites including South Africa, The Gambia, Uganda and Ethiopia. In brief, when a newly diagnosed TB case was identified, individuals with whom they shared a house for a minimum period of three months were recruited with the expectation that they would have been exposed (likely repeatedly) and infected with *M.tb* and were, therefore, at high risk of developing TB. A total of 4,466 household contacts were followed for two years and whole blood was collected at recruitment (time zero), 6 and 18 months thereafter for RNAseq. Further details of the study details have been described previously [22, 23]. The samples have been used in several previous publications for validation and discovery of TB biomarkers [22, 23]. The raw RNAseq FASTQ files for the study were downloaded from the GEO public database (accession number GSE94438).

The second data set used included a subset of RNAseq data from the South African Catalysis study (GEO: GSE89403) which was a longitudinal study of resolution of lung inflammation in TB cases. Here, we only used RNAseq data from TB patients at diagnosis and the non-TB controls [27, 28] to enlarge the size of the sample of TB at diagnosis patients. The study

recruited adult HIV-negative TB patients, whose diagnosis was confirmed with a sputum culture [27, 28]. Asymptomatic individuals that were recruited from the same community and tested negative for TB on sputum and chest X-ray were used as controls. The results of 18-F fluorodeoxyglucose (FDG) PET-CT scans from TB patients at enrollment were used to categorize patients into groups with high and low levels of lung inflammation activity based on PET. The metric used is a sum of total glycolytic activity index (TGAI) [27, 70] of all metabolically active lesions and the products of the mean lesion intensities with the cavity volumes, abbreviated ComTGAI, and is correlated with levels of lung inflammation [27, 71] Patients were dichotomized into low and high PET scores based on a threshold of 4,000 units at baseline.

## Ethics statement

Both source studies (the GC6-74 study and the Catalysis TB Biomarker study) were performed with ethical approvals and required written informed consent. For the GC6-74 study the following ethics approvals applied (as described in [23]): Stellenbosch University, South Africa, Stellenbosch University Human Research Ethics Committee, N05/11/187; UK Medical Research Council Unit, the Gambia, Joint Medical Research Council and Gambian Government, SCC.1141vs2; Makarere University, Uganda, Uganda National Council for Science and Technology, MV 715, and University Hospitals Case Medical Centre, 12-95-08; Armauer Hansen Research Institute, Ethiopia, Armauer Hansen Research Institute (AHRI)/All Africa Leprosy, TB and Rehabilitation Training Center (ALERT), P015/10; and the University of Cape Town, South Africa, University of Cape Town Human Research Ethics Committee (HREC), 013/2013. For the Catalysis TB Biomarker study, the ethical approval was from the Stellenbosch University, South Africa, Stellenbosch University Human Research Ethics Committee, N10/01/013 (as described in [28]).

This study was performed with the following ethical approvals that are still current: Stellenbosch University, South Africa, Stellenbosch University Human Research Ethics Committee, N05/11/187 and Stellenbosch University, South Africa, Stellenbosch University Human Research Ethics Committee, N10/01/013.

## RNAseq and quality control

The data used in the current study had been obtained with RNA extracted from whole blood and sequenced on the Illumina HiSeq-4000 (GC6-74) and HiSeq-2000 (Catalysis) platforms generating 50 bp stranded paired-end reads. We used the FastQC program (version 0.11.5) [72] to assess the quality of the reads.

## Read mapping

The Spliced Transcripts Alignment to a Reference (STAR) software (version STAR_2.5.3a) [73] was used to map reads to the Ensembl [74] human genome primary assembly (version GRCh38.89). The quantMode GeneCounts option was selected to generate raw genewise read counts for each sample.

## Differential expression analysis

The DE analysis was performed in R [75] using the edgeR (version 3.26.8) [76] Bioconductor [77] package. Briefly, raw counts were filtered to remove genes with low expression, normalized, and negative binomial generalized linear models were fitted. In addition to time before diagnosis, sex and site were included as factors in the model matrix since they were observed

to be responsible for the separation of principal component (PC)1 and PC2 based on multidimensional scaling plots.

The linear model applied was:

$$Gene = 0 + Time + sex + site$$

In R syntax as: model.matrix(~ 0 +TimeBf+ sex + site), i.e., a model without an intercept

The quasi-likelihood F-test (QLF-Test) was used to determine significance and identify DE genes. For the GC6-74 data set, all TB progressor groups were compared to the non-TB baseline group (time 0). For the Catalysis data set, the TB patients were compared to non-TB controls. TB patients categorized with high and low PET scores were also separately compared to controls and directly to each other to identify differences. Significantly DE genes were selected that had a false discovery rate (FDR) <0.05.

To determine the influence that the larger sample sizes [1–6 months before (n = 47) and TB (n = 90) compared to 13–18 months before (n = 18) and 7–12 months before (n = 19), see Table 1] had on the number of DE genes detected, the 1–6 months before diagnosis and TB groups with larger samples were subsampled. These groups were subsampled 1000 times at a sample size of 19 and a DE analysis was performed for each iteration.

## Gene ontology and KEGG pathway analysis

The topGO [78] Bioconductor package was used to test for enrichments in any GO terms [79] associated with the DE genes. The GO graph structure was generated using both the "classic" and "weight01", algorithms using the Fisher's exact test to identify enriched terms. In brief, the classic algorithm tests each GO category independently. The "weight01" is the default algorithm used by the topGO package and essentially penalizes scores for more general terms that share genes with more specific neighboring terms, weighting the analysis for the identification of more specific and therefore informative ontologies. "Weight01" is a mixture of two algorithms, "elim" which eliminates genes shared between a node and its ancestor, and "weight" which weights genes locally as a function of the ratio between child and ancestor nodes [78]. The weight01 mixture of the two algorithms moderates the extremes of the two individual algorithms.

The kegga function in edgeR was used to perform a KEGG [80] pathway enrichment analysis on the DE gene sets. The Bioconductor [77] pathview package [29] was subsequently used to visualize gene expression on enriched KEGG pathway graphs.

Due to the extensive overlap of genes in the hierarchical GO categories, neither the topGO or kegga programs correct for multiple hypothesis testing and recommend against it. We therefore applied a stringent p-value cut-off of $<5 \times 10^{-4}$ to identify enriched pathways.

## Supporting information

**S1 Table. PET scores for the catalysis study specimens.** The PET score (low-PET, high-PET as defined in Methods) for the Catalysis study specimens used in the current study. (XLSX)

**S2 Table. List of differentially expressed genes at all time points.** The statistical metrics presented for each comparison include: log2 fold change (log2FC), average log2 counts per million (logCPM), quasi-likelihood F-statistic (F), p-value (PValue) and false discovery rate (FDR). The non-TB time zero baseline group from the GC6 study were the control group for all comparisons prior to TB diagnosis (i.e., 24, 18, 12 and 6 months before diagnosis) while the

Catalysis study healthy controls served as the baseline for the TB comparison.
(XLSX)

**S3 Table. Gene ontology (GO) enrichment analysis results for significantly upregulated genes at all time points.** The analysis was performed using the topGO R Bioconductor package. The statistical metrics presented for each process include: the total number of genes annotated to the process (Total annotated), the number of genes that were significantly up (N up) -regulated, the percent of total annotated that were up-regulated (% Up) and the number that were expected by chance (Expected). The uncorrected Fisher's exact test p-value (Pvalue) and overall rank for over-representation of the GO term in the set using both the classic and weight01 (W1) algorithms are presented.
(XLSX)

**S4 Table. KEGG pathway enrichment analysis results for significantly differentially expressed genes at all time points.** The analysis was performed using the kegga function from the edgeR R Bioconductor package. The statistical metrics presented for each pathway include: the total number of genes annotated to the pathway (Total annotated), the number of genes that were significantly up (N Up) or down (N Down) -regulated, the % of the total annotated that were up (% Up) or down (% Down) -regulated and the uncorrected Fisher's exact test p-value (Pvalue) for up and down comparisons (P Up, P Down) for over-representation of the KEGG pathway in the set.
(XLSX)

**S5 Table. KEGG and gene ontology (GO) enrichment analyses for significantly differentially expressed genes between low and high PET score groups for the terms and pathways identified in the main analyses.** The analyses were performed using the topGO and kegga function from the edgeR R Bioconductor packages. The statistical metrics presented are as for S2 and S3 Tables respectively.
(XLSX)

**S1 Fig. Density plot of subsampling the 1–6 months before TB diagnosis sample numbers.** Subsampling was performed on the 1–6 month-before TB group to investigate the effect the larger sample size (n = 47) had on the number of significantly up-regulated genes identified. A total of 1000 differential expression analysis tests were performed using a sample size of 19 which is similar to the earlier 12 month before (n = 19) and 18-month before time points (n = 18). The mean number of significantly up-regulated genes identified in the subsampling was 1,436 (95% CI:120, 3,100). This number was lower than the 2,791 identified with the full 47 samples but far higher than the number identified in the 18 (249; n = 19) and 12 (492; n = 18) months before diagnosis groups.
(TIF)

**S2 Fig. Density plot of subsampling the TB group sample numbers.** Subsampling was performed on the Catalysis study TB group to investigate the effect the larger sample size (n = 90) had on the number of significantly up-regulated genes identified. A total of 1000 differential expression analysis tests were performed using a sample size of 19 which is similar to the earlier 12 month before (n = 19) and 18-month before time points (n = 18). The mean number of significantly upregulated genes identified in the subsampling was 3,485 (95% CI:2,719, 4,190). This was lower than the 4,391 identified with the full 90 samples but far higher than the number identified in the 18 (249; n = 19) and 12 (492; n = 18) months before diagnosis groups.
(TIF)

## Acknowledgments

The authors acknowledge the Centre for High Performance Computing (CHPC), South Africa, for providing computational resources to this research project.

Members of The Bill and Melinda Gates Foundation Grand Challenges 6, 74 (GC6-74) "Biomarkers of Protective Immunity Against TB in the Context of HIV/AIDS in Africa" Consortium: Gerhard Walzl[1,2,3], Gillian F. Black[1,2,3], Gian van der Spuy[1,2,3,4], Kim Stanley[1,2,3], Magdalena Kriel[1,2,3], Nelita Du Plessis[1,2,3], Nonhlanhla Nene[1,2,3], Teri Roberts[1,2,3], Léanie Kleynhans[1,2,3], Andrea Gutschmidt[1,2,3], Bronwyn Smith[1,2,3], Andre G. Loxton[1,2,3], Novel N. Chegou[1,2,3], Gerard Tromp[1,2,3,4,5], David Tabb[1,2,3,4,5], Tom H. M. Ottenhoff[6], Michel R. Klein[6], Marielle C. Haks[6], Kees L. M. C. Franken[6], Annemieke Geluk[6], Krista E. van Meijgaarden[6], Simone A. Joosten[6], W. Henry Boom[7], Bonnie Thiel[7], Harriet Mayanja-Kizza[8], Moses Joloba[8], Sarah Zalwango[8], Mary Nsereko[8], Brenda Okwera[8], Hussein Kisingo[8], Stefan H. E. Kaufmann[9] (GC6-74 principal investigator), Shreemanta K. Parida[9], Robert Golinski[9], Jeroen Maertzdorf[9], January Weiner 3rd[9], Marc Jacobson[9], Hazel M. Dockrell[10], Maeve Lalor[10], Steven Smith[10], Patricia Gorak-Stolinska[10], Yun-Gyoung Hur[10], Ji-Sook Lee[10], Amelia C. Crampin[11], Neil French[11], Bagrey Ngwira[11], Anne Ben-Smith[11], Kate Watkins[11], Lyn Ambrose[11], Felanji Simukonda[11], Hazzie Mvula[11], Femia Chilongo[11], Jacky Saul[11], Keith Branson[11], Sara Suliman[12], Thomas J. Scriba[12], Hassan Mahomed[12], E. Jane Hughes[12], Nicole Bilek[12], Mzwandile Erasmus[12], Onke Xasa[12], Ashley Veldsman[12], Katrina Downing[12], Michelle Fisher[12], Adam Penn-Nicholson[12], Humphrey Mulenga[12], Brian Abel[12], Mark Bowmaker[12], Benjamin Kagina[12], William Kwong Chung[12], Willem A. Hanekom[12], Jerry Sadoff[13], Donata Sizemore[13], S. Ramachandran[13], Lew Barker[13], Michael Brennan[13], Frank Weichold[13], Stefanie Muller[13], Larry Geiter[13], Desta Kassa[14], Almaz Abebe[14], Tsehayenesh Mesele[14], Belete Tegbaru[14], Debbie van Baarle[15], Frank Miedema[15], Rawleigh Howe[16], Adane Mihret[16], Abraham Aseffa[16], Yonas Bekele[16], Rachel Iwnetu[16], Mesfin Tafesse[16], Lawrence Yamuah[16], Martin Ota[17], Jayne Sutherland[17], Philip Hill[17], Richard Adegbola[17], Tumani Corrah[17], Martin Antonio[17], Toyin Togun[17], Ifedayo Adetifa[17], Simon Donkor[17], Peter Andersen[18], Ida Rosenkrands[18], Mark Doherty[18], Karin Weldingh[18], Gary Schoolnik[19], Gregory Dolganov[19], and Tran Van[19]

[1.] Division of Molecular Biology and Human Genetics, Department of Biomedical Sciences, Stellenbosch University, Cape Town, South Africa

[2.] Department of Science and Innovation/National Research Foundation Centre of Excellence for Biomedical Tuberculosis Research, Stellenbosch University, Cape Town, South Africa

[3.] South African Medical Research Council Centre for Tuberculosis Research, Stellenbosch University, Cape Town, South Africa

[4.] South African Tuberculosis Bioinformatics Initiative, Stellenbosch University, Cape Town, South Africa

[5.] Centre for Bioinformatics and Computational Biology, Stellenbosch University, Stellenbosch, South Africa

[6.] Department of Infectious Diseases, Leiden University Medical Centre, Leiden, the Netherlands

[7.] Tuberculosis Research Unit, Department of Medicine, Case Western Reserve University School of Medicine and University Hospitals Case Medical Center, Cleveland, Ohio, USA

[8.] Department of Medicine and Department of Microbiology, College of Health Sciences, Faculty of Medicine, Makerere University, Kampala, Uganda

[9.] Department of Immunology, Max Planck Institute for Infection Biology, Berlin, Germany

10. Department of Immunology and Infection, Faculty of Infectious and Tropical Diseases, London School of Hygiene and Tropical Medicine, London, United Kingdom

11. Karonga Prevention Study, Chilumba, Malawi

12. South African Tuberculosis Vaccine Initiative, Institute of Infectious Disease and Molecular Medicine and Division of Immunology, Department of Pathology, University of Cape Town, Cape Town, South Africa

13. Aeras, Rockville, Maryland, USA

14. Ethiopian Health and Nutrition Research Institute, Addis Ababa, Ethiopia

15. University Medical Centre, Utrecht, the Netherlands

16. Armauer Hansen Research Institute, Addis Ababa, Ethiopia

17. Vaccines and Immunity Theme, Medical Research Council Unit, Fajara, the Gambia

18. Department of Infectious Disease Immunology, Statens Serum Institute, Copenhagen, Denmark

19. Department of Microbiology and Immunology, Stanford University, Stanford, California, USA

Members of The Catalysis Biomarker Consortium: Stephanus T. Malherbe[1,2], Patrick Dupont[3,4], Ilse Kant[4], Katharina Ronacher[1,2], Magdalena Kriel[1,2], André G. Loxton[1,2], Ray Y. Chen[5], Laura E. Via[5,6], Friedrich Thienemann[6,7], Robert J. Wilkinson[6,7,8,9], Clifton E. Barry III[1,2,5,6], Stephanie Griffith-Richards[10], Annare Ellman[4], Jill Winter[11], Gerhard Walzl[1,2], Nelita Du Plessis[1,2], Caroline G.G. Beltran[1,2], Lani Thiart[1,2], Gerard Tromp[1,2], Lance A. Lucas[1,2], Bronwyn Smith[1,2], Kim Stanley[1,2], David Alland[12], Shubhada Shenai[12], Lori E. Dodd[13], and James M. Warwick[4].

1. Department of Science and Innovation/National Research Foundation Centre of Excellence for Biomedical Tuberculosis Research and South African Medical Research Council Centre for Tuberculosis Research.

2. Division of Molecular Biology and Human Genetics, Faculty of Medicine and Health Sciences, Stellenbosch University, Cape Town, South Africa.

3. Laboratory for Cognitive Neurology, Department of Neurosciences, Katolieke Universiteit, Leuven, Belgium.

4. Division of Nuclear Medicine, Department of Medical Imaging and Clinical Oncology, Faculty of Medicine and Health Sciences, Stellenbosch University, Cape Town, South Africa.

5. Tuberculosis Research Section, Laboratory of Clinical Infectious Diseases, Division of Intramural Research, National Institute of Allergy and Infectious Diseases, National Institutes of Health, Bethesda, Maryland, USA.

6. Wellcome Centre for Infectious Diseases Research in Africa, Institute of Infectious Disease and Molecular Medicine, Faculty of Health Science, University of Cape Town, Cape Town, South Africa

7. Department of Medicine, Groote Schuur Hospital, Faculty of Health Science, University of Cape Town, Cape Town, South Africa.

8. The Francis Crick Institute, London, United Kingdom

9. Department of Medicine, Imperial College London, London, United Kingdom

10. Division of Radiodiagnosis, Department of Medical Imaging and Clinical Oncology, Faculty of Medicine and Health Sciences, Stellenbosch University, Cape Town, South Africa.

11. Catalysis Foundation for Health, Emeryville, California, USA.

12. Center for Emerging Pathogens, Department of Medicine, Rutgers-New Jersey Medical School, Rutgers Biomedical and Health Sciences, Newark, New Jersey, USA.

13. Biostatistics Research Branch, National Institute of Allergy and Infectious Diseases, National Institutes of Health Bethesda, Maryland, USA.

## Author Contributions

**Conceptualization:** Stuart Meier, James A. Seddon, Gerhard Walzl, Gerard Tromp.

**Data curation:** Stuart Meier, Stephanus T. Malherbe, Daniel E. Zak, Ethan Thompson.

**Formal analysis:** Stuart Meier, Gerard Tromp.

**Funding acquisition:** Stefan H. E. Kaufmann, Tom H. M. Ottenhoff, Jill Winter, Gerhard Walzl.

**Investigation:** Elizna Maasdorp, Léanie Kleynhans, Nelita du Plessis, Andre G. Loxton, Stephanus T. Malherbe, Stefan H. E. Kaufmann, Thomas J. Scriba, Jayne S. Sutherland, Jill Winter.

**Project administration:** Stefan H. E. Kaufmann, Tom H. M. Ottenhoff, Jayne S. Sutherland, Jill Winter, Gerhard Walzl.

**Supervision:** Gerhard Walzl, Gerard Tromp.

**Visualization:** Stuart Meier, Gerard Tromp.

**Writing – original draft:** Stuart Meier, James A. Seddon, Helena Kuivaniemi, Gerard Tromp.

**Writing – review & editing:** Stuart Meier, James A. Seddon, Elizna Maasdorp, Léanie Kleynhans, Nelita du Plessis, Andre G. Loxton, Stephanus T. Malherbe, Daniel E. Zak, Ethan Thompson, Fergal J. Duffy, Stefan H. E. Kaufmann, Tom H. M. Ottenhoff, Thomas J. Scriba, Sara Suliman, Jayne S. Sutherland, Jill Winter, Helena Kuivaniemi, Gerhard Walzl, Gerard Tromp.

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
