## [Decision Letter · Decision Letter 0]

13 Sep 2022

PONE-D-22-18474Neutrophil degranulation, NETosis and platelet degranulation pathway genes are co-induced in whole blood up to six months before tuberculosis diagnosisPLOS ONE

Dear Dr. Tromp,

Thank you for submitting your manuscript to PLOS ONE. After careful consideration, we feel that it has merit but does not fully meet PLOS ONE’s publication criteria as it currently stands. Therefore, we invite you to submit a revised version of the manuscript that addresses the points raised during the review process.

We look forward to receiving your revised manuscript.

Kind regards,

Srinivas Mummidi, D.V.M., Ph.D.

Academic Editor

PLOS ONE

Journal Requirements:

Reviewers' comments:

Reviewer's Responses to Questions

**Comments to the Author**

1. Is the manuscript technically sound, and do the data support the conclusions?

Reviewer #1: Yes

2. Has the statistical analysis been performed appropriately and rigorously? 

Reviewer #1: Yes

3. Have the authors made all data underlying the findings in their manuscript fully available?

Reviewer #1: Yes

4. Is the manuscript presented in an intelligible fashion and written in standard English?

Reviewer #1: Yes

5. Review Comments to the Author

Reviewer #1: This manuscript by Stuart Meier et al, is a well designed and well conceived study and identifies some of the key host signatures of early diagnosis of TB. Through RNAseq analysis of whole blood and PET-CT imaging in 2 different cohorts spanning early stage of infection to TB, the study provides key insights into the probable pathways that could be engaged in TB pathogenesis during progression of the disease. Through this analysis, the authors show clear correlation between activation of neutrophils, platelet degranulation and NET formation, and progression of disease over a period of 18 months. Overall the study has very high scientific merit and advances our understanding about the key components of the host that could be involved in TB immunopathogenesis. Some of the minor comments, if addressed can help improve this manuscript further:

1. One of the sub division of cohorts could be house hold contacts that developed TB vs those that never developed TB. If the analysis can include these sub cohorts, it can provide more comprehensive evidence of the claimed findings.

2. It is not clear from this manuscript if PET-CT imaging of TB progresser's was done and compared with house hold contacts that never developed TB. Perhaps a PET-CT from this cohort can be further divided into LTBI positive vs LTBI negative to better predict if PET-CT data hold any true relevance.

3. The discussion can provide more insight about the relationship between Type 1 IFN signaling and Neutrophil activation/NETOsis pathways.

4. Primary TB can be infectious as has been shown in earlier studies, whereas authors here suggest otherwise (Line 387 discussion section).

5. Figure 3 is completely blur and not readable at all.

6. PLOS authors have the option to publish the peer review history of their article (what does this mean?). If published, this will include your full peer review and any attached files.

Reviewer #1: **Yes: **Arshad Khan

---

## [Author Response · Author response to Decision Letter 0]

18 Oct 2022

Each comment is completely addressed in the Response to Reviewers document.

Editorial staff

1. Please ensure that your manuscript meets PLOS ONE's style requirements, including those for file naming. -- Checked and used a PLoS Template.

2. We note that you have included the phrase “data not shown” in your manuscript. -- Replaced with the relevant information.

3. Please include your full ethics statement in the ‘Methods’ section of your manuscript file. -- Relevant ethics statement included with the ethics review numbers.

4. Please review your reference list to ensure that it is complete and correct. -- Done. Included description of process used to check.

Reviewer.

1. One of the sub division of cohorts could be house hold contacts that developed TB vs those that never developed TB. -- Edited to respond to critique.

2. It is not clear from this manuscript if PET-CT imaging of TB progresser's was done and compared with house-hold contacts that never developed TB. -- Responded. Data were not available.

3. The discussion can provide more insight about the relationship between Type 1 IFN signaling and Neutrophil activation/NETOsis pathways. -- Extensive rewrite of relevant section

4. Primary TB can be infectious as has been shown in earlier studies, whereas authors here suggest otherwise (Line 387 discussion section). -- Edits to correct as suggested. 

5. Figure 3 is completely blur and not readable at all. -- We regret that reviewer did not have access to high-resolution figures. The submitted figures were high quality, but editorial management system reduced resolution. Willing to provide high-resolution figures or request editorial staff to make them available. 

Details as provided in the response to reviewers document.

COMMENTS FROM EDITORIAL TEAM:

and 

RESPONSE: 

We have checked the style requirements documents and used the Word Template. We have ensured that the formatting was not altered from the format in the template. 

RESPONSE: 

The phrase “data not shown” has been removed and the paragraph updated to the following:

Mapping statistics revealed that for the GC6-74 cohort, a mean of 47.5 million reads [(47.1, 47.9: 95% CI; 85.9% (95% CI: 85.7, 86.1)] mapped uniquely to the human genome with 87.8 % (95% CI: 87.7, 87.9) of these mapping to genes. For the Catalysis cohort, a mean of 40.1 million reads [(39.4, 40.8: 95% CI; 88.6% (88.3, 88.9: 95% CI)] mapped uniquely to the human genome with 88.8 % (95% CI: 88.7, 88.9) mapping to genes.

RESPONSE: 

We used data deposited in public repositories that had previously been collected with ethical approval and informed consent. In the revised submission, we now list all the prior ethics approvals under which the data were collected.

In paper:

Further details of the study, including ethical approval details have been described previously [23].

For the GC6-74 study [23] the reported ethics approvals are as follows (from one of the appendices):

Site Ethics Review Committee Protocol no.

SUN Stellenbosch University Institutional Review Board N05/11/187

MRC Joint Medical Research Council and Gambian Government SCC.1141vs2

MAK Uganda National Council for Science and Technology MV 715

MAK University Hospitals Case Medical Centre 12-95-08

AHRI Armauer Hansen Research Institute (AHRI)/All Africa Leprosy, TB and Rehabilitation Training Center (ALERT) P015/10

UCT (GC6) University of Cape Town Human Research Ethics Committee HREC 013/2013

For the Catalysis study [28], the ethical approval (single site) was as follows:

SU Stellenbosch University Institutional Review Board N10/01/013

We have inserted the following in the Methods section at the end of the sub-section titled “Study outline and data sources”

Both source studies (the GC6-74 study and the Catalysis TB Biomarker study) were performed with ethical approvals and required written informed consent. For the GC6-74 study the following ethics approvals applied (as described in [23]): Stellenbosch University, South Africa, Stellenbosch University Institutional Review Board, N05/11/187; UK Medical Research Council Unit, the Gambia, Joint Medical Research Council and Gambian Government, SCC.1141vs2; Makarere University, Uganda, Uganda National Council for Science and Technology, MV 715, and University Hospitals Case Medical Centre, 12-95-08; Armauer Hansen Research Institute, Ethiopia, Armauer Hansen Research Institute (AHRI)/All Africa Leprosy, TB and Rehabilitation Training Center (ALERT), P015/10; and the University of Cape Town, South Africa, University of Cape Town Human Research Ethics Committee (HREC), 013/2013. For the Catalysis TB Biomarker study, the ethical approval was from the Stellenbosch University, South Africa, Stellenbosch University Institutional Review Board, N10/01/013 [28].

RESPONSE: 

We performed a thorough review of all publications cited (on October 2, 2022). None of the citations were retracted. There were however two corrections (but these are indicated as such on the publisher site and are corrected in the online published versions), and 19 comments (indicated on the publisher web site and on PubMed).

 

COMMENTS FROM REVIEWER Arshad Khan

This manuscript by Stuart Meier et al, is a well designed and well conceived study and identifies some of the key host signatures of early diagnosis of TB. Through RNAseq analysis of whole blood and PET-CT imaging in 2 different cohorts spanning early stage of infection to TB, the study provides key insights into the probable pathways that could be engaged in TB pathogenesis during progression of the disease. Through this analysis, the authors show clear correlation between activation of neutrophils, platelet degranulation and NET formation, and progression of disease over a period of 18 months. Overall the study has very high scientific merit and advances our understanding about the key components of the host that could be involved in TB immunopathogenesis. 

RESPONSE: 

We thank the reviewer for the constructive comments, and outline responses to specific comments below.

Some of the minor comments, if addressed can help improve this manuscript further: 

1. One of the sub division of cohorts could be house hold contacts that developed TB vs those that never developed TB. If the analysis can include these sub cohorts, it can provide more comprehensive evidence of the claimed findings.

RESPONSE: 

The reviewer is correct that a purified set of controls of individuals resistant to TB (never developed TB) would be more powerful for detecting the difference between TB cases and controls. Unfortunately, the design of the GC6-74 study did not permit detecting such a sub-cohort. The analysis of the GC6-74 progressor cohort compared individuals that developed TB to those who did not during the period of 24 months of follow-up. The subjects in this study were all household contacts of recently diagnosed TB cases. Subjects who developed TB after 3 months (washout) and within two years of enrollment in the study were classified as TB progressors while those who did not were the control group. Most cases developed TB within a year with a median of 9 months and an IQR (6,17). Determining that a sub-cohort of the 4,362 subjects who did not develop TB by 24 months never developed TB would be an enormous task both in terms of labor and cost, and would be extremely difficult in the high prevalence of TB in each of the regions from which the cohort was derived. 

The potential of undetected TB, either latent or incipient, among the controls, would however only decrease the power to detect the effect and not fundamentally alter the observations. One could phrase it to say that the observations are more robust since the controls were not highly selected. Certainly, more selected, or purified, controls could have provided insight on aspects of the analysis that could have been obscured by the potential presence of disease signals among the controls.

We have added the following sentence at the end of the first paragraph under the Study groups and data quality sub-section of Results. 

Due to the finite follow-up some of the subjects classified as non-progressor might have had pre-clinical TB. Such subjects in the “control” group would dilute the disease signals, possibly obscuring some insights, but would not alter the fundamental findings.

2. It is not clear from this manuscript if PET-CT imaging of TB progresser's was done and compared with house-hold contacts that never developed TB. Perhaps a PET-CT from this cohort can be further divided into LTBI positive vs LTBI negative to better predict if PET-CT data hold any true relevance.

RESPONSE: 

The PET-CT imaging was only performed on the Catalysis TB cohort (a longitudinal treatment response study). Unfortunately, it was not available for the GC6 progressor cohort (a prospective longitudinal follow-up study).

3. The discussion can provide more insight about the relationship between Type 1 IFN signaling and Neutrophil activation/NETOsis pathways. 

RESPONSE: 

Thank you for this important comment. We have revised the relevant paragraph in the Discussion about type I IFN, as below.

Submitted manuscript:

Given the early and sustained level of type I IFN signaling observed during TB progression, it is interesting to note that IFN-α can prime neutrophils for NETosis [58]. In turn, NETs can stimulate type I IFN production in plasmacytoid dendritic cells (pDCs), prolonging the inflammatory response [59]. 

Revised manuscript:

Given the early and sustained level of type I IFN signaling observed during TB progression, it is interesting to note that in TB susceptible mice, type I IFN signaling has been shown to induce NETosis through activation of interferon α and β receptor subunit 1, which is associated with enhanced mycobacterial growth at infection sites and enhanced TB pathogenesis [58]. The same study identified NETs in nectrotic lung lesions of TB patients that responded poorly to treatment. Further, serum from patients with autoimmune disorders that have elevated levels of type I IFNs, as well as exogenous IFN-α, has been shown to stimulate neutrophil NET production in vitro suggesting that type I IFN prime neutrophils for NET production [59-61]. In turn, self-DNA and antimicrobial peptides released with NETs, have been reported to induce the chronic activation of plasmacytoid dendritic cells and secretion of type I IFNs in SLE patients creating a positive feedback loop that prolongs the inflammatory response [62]. The upregulated type I IFN signaling observed in this study that precedes neutrophil activation and NETosis, along with the above-mentioned studies, is consistent with type I IFN signaling diving neutrophil activation and NETosis during TB progression and enhancing TB pathogenesis.

4. Primary TB can be infectious as has been shown in earlier studies, whereas authors here suggest otherwise (Line 387 discussion section).

RESPONSE: 

We thank the reviewer for detecting this inference that was not intended. We have revised the manuscript as below.

Submitted manuscript: (1st para of discussion)

Our results revealed an early induction of IFN-related signaling at 18 months before diagnosis. A strong induction of neutrophil and platelet degranulation and NETosis related genes was detected 6 months before TB diagnosis and persisted in TB patients supporting a pathogenic role of these responses in disease development. Although these processes have previously been detected in severe TB, here we document that they occur well before disease diagnosis, indicating that they may be critical in mediating lung tissue destruction and progression from non-infectious latent or primary TB to contagious TB. 

Revised manuscript:

Our results revealed an early induction of IFN-related signaling at 18 months before diagnosis. A strong induction of neutrophil and platelet degranulation and NETosis related genes was detected 6 months before TB diagnosis and persisted in TB patients supporting a pathogenic role of these responses in disease development. Although these processes have previously been detected in patients diagnosed with active TB, here we document that they occur well before diagnosis, indicating that they may be critical in mediating lung tissue destruction and progression from infection with M.tb to active TB.

5. Figure 3 is completely blur and not readable at all.

RESPONSE: 

We apologize to the reviewer that the image quality in the review document was poor. It appears that the PLOS One submission system reduces the quality of the images, possibly to keep the size of the submitted document down to a reasonable magnitude. We agree that the quality of the figures as produced by the PLOS One online submission system is sub-optimal. The submitted figures were high-quality and we would be happy to share these with the reviewer. Perhaps the editorial system can provide the reviewer access to the original high-quality figures?

---

## [Decision Letter · Decision Letter 1]

15 Nov 2022

Neutrophil degranulation, NETosis and platelet degranulation pathway genes are co-induced in whole blood up to six months before tuberculosis diagnosis

PONE-D-22-18474R1

Dear Dr. Tromp,

We’re pleased to inform you that your manuscript has been judged scientifically suitable for publication and will be formally accepted for publication once it meets all outstanding technical requirements.

Kind regards,

Srinivas Mummidi, D.V.M., Ph.D.

Academic Editor

PLOS ONE

Additional Editor Comments (optional):

Reviewers' comments:

Reviewer's Responses to Questions

**Comments to the Author**

1. If the authors have adequately addressed your comments raised in a previous round of review and you feel that this manuscript is now acceptable for publication, you may indicate that here to bypass the “Comments to the Author” section, enter your conflict of interest statement in the “Confidential to Editor” section, and submit your "Accept" recommendation.

Reviewer #1: All comments have been addressed

2. Is the manuscript technically sound, and do the data support the conclusions?

Reviewer #1: Yes

3. Has the statistical analysis been performed appropriately and rigorously? 

Reviewer #1: Yes

4. Have the authors made all data underlying the findings in their manuscript fully available?

Reviewer #1: Yes

5. Is the manuscript presented in an intelligible fashion and written in standard English?

Reviewer #1: Yes

6. Review Comments to the Author

Reviewer #1: The authors have addressed the comments raised by reviewer and the manuscript can now be accepted for publication.

7. PLOS authors have the option to publish the peer review history of their article (what does this mean?). If published, this will include your full peer review and any attached files.

Reviewer #1: **Yes: **Arshad Khan

---

## [Editor Report · Acceptance letter]

21 Nov 2022

PONE-D-22-18474R1 

Neutrophil degranulation, NETosis and platelet degranulation pathway genes are co-induced in whole blood up to six months before tuberculosis diagnosis 

Dear Dr. Tromp:

I'm pleased to inform you that your manuscript has been deemed suitable for publication in PLOS ONE. Congratulations! Your manuscript is now with our production department. 

Kind regards, 

on behalf of

Srinivas Mummidi 

Academic Editor

PLOS ONE